# Modeling historic incidence trends implies early field cancerization in esophageal squamous cell carcinoma

**Georg E. Luebeck**[1]*, **Thomas L. Vaughan**[2], **Kit Curtius**[3], **William D. Hazelton**[1]

1 Public Health Sciences Division, Computational Biology Program, Fred Hutchinson Cancer Research Center, Seattle, Washington, United States of America, 2 Professor Emeritus, Public Health Sciences Division, Cancer Epidemiology Program, Fred Hutchinson Cancer Research Center, Seattle, Washington, United States of America, 3 Division of Biomedical Informatics, Department of Medicine, University of California, San Diego, La Jolla, California, United States of America

* gluebeck@fredhutch.org

**Data Availability Statement:** The data underlying the results presented in the study are available from https://seer.cancer.gov/data/access.html.

## Abstract

Patterns of cancer incidence, viewed over extended time periods, reveal important aspects of multistage carcinogenesis. Here we show how a multistage clonal expansion (MSCE) model for cancer can be harnessed to identify biological processes that shape the surprisingly dynamic and disparate incidence patterns of esophageal squamous cell carcinoma (ESCC) in the US population. While the dramatic rise in esophageal adenocarcinoma (EAC) in the US has been largely attributed to reflux related increases in the prevalence of Barrett's esophagus (BE), the premalignant field in which most EAC are thought to arise, only scant evidence exists for field cancerization contributing to ESCC. Our analyses of incidence patterns suggest that ESCC is associated with a premalignant field that may develop very early in life. Although the risk of ESCC, which is substantially higher in Blacks than Whites, is generally assumed to be associated with late-childhood and adult exposures to carcinogens, such as from tobacco smoking, alcohol consumption and various industrial exposures, the temporal trends we identify for ESCC suggest an onset distribution of field-defects before age 10, most strongly among Blacks. These trends differ significantly in shape and strength from field-defect trends that we estimate for US Whites. Moreover, the rates of ESCC-predisposing field-defects predicted by the model for cohorts of black children are decreasing for more recent birth cohorts (for Blacks born after 1940). These results point to a potential etiologic role of factors acting early in life, perhaps related to nutritional deficiencies, in the development of ESCC and its predisposing field-defect. Such factors may explain some of the striking racial differences seen in ESCC incidence patterns over time in the US.

## Author summary

We used a cell-level carcinogenesis model to analyze incidence patterns of esophageal squamous cell carcinoma (ESCC) in the US. We found an important role of an esophageal field-defect that is predicted to occur predominantly in childhood and predisposes to

**Funding:** This research was supported by the National Cancer Institute (www.cancer.gov) grant U01CA152926 (to GEL, TLV, WDH) and Moores Cancer Center Support Grant NCI P30 CA023100 (to KC). The funders had no role in study design, data collection and analysis, decision to publish, or preparation of the manuscript.

**Competing interests:** The authors have declared that no competing interests exist.

ESCC in adult life. Age-specific ESCC incidence patterns are also known to differ considerably between Blacks and Whites, and between males and females in the US, but the model consistently predicts early-childhood field-defects in all four groups. The estimated historical field-defect trends appear consistent with possible early childhood nutritional deficiencies.

## Introduction

Esophageal squamous cell carcinoma (ESCC) is the main histologic type of esophageal cancer worldwide and remains a significant cause of cancer morbidity and mortality [1]. In the United States and other parts of the Western world, however, the incidence of ESCC has significantly decreased in the past 3 to 4 decades among virtually every race/ethnicity [1, 2], and is now dominated by the incidence of esophageal adenocarcinoma (EAC) which continues to rise in Western countries for reasons that are still not fully understood [3, 4]. In spite of the opposing incidence trends for ESCC and EAC, nutritional deficiencies and cigarette smoking are significant risk factors for both histologic subtypes [5–7]. However, excess alcohol consumption is significantly associated with increased risk for ESCC (but weakly or inconsistently associated with EAC), while abdominal obesity and gastro-esophageal reflux are prominent risk factors for Barrett's esophagus (BE) and EAC [4–9]. BE develops in the lower esophagus and is a metaplastic premalignant field that is generally considered to be a requisite precursor for progression to EAC [10]. In contrast, the known precursors for ESCC (mild and severe esophageal dysplasia) appear to arise within normal appearing squamous tissue [11]. However, the high frequency of metachronous presentation suggests an underlying predisposition or tissue field-defect that significantly increases the risk of dysplastic lesions prior to the occurrence of ESCC [12–14].

Here we present a novel computational method to 1) disentangle age, period and cohort (APC) effects that modulate the incidence of ESCC under the constraints imposed by a stochastic carcinogenesis model for ESCC and 2) explore the impact of cohort and historical trends on the initial mutational events that predispose to ESCC. Specifically, our method imposes smoothed period and cohort trends on the biological model parameters that control the initiation of a premalignant field-defect, as well as subsequent clonal expansions that may arise within the field. Our approach differs from traditional statistical APC approaches (multiplicative constants) in that period effects coincide with continuous changes in environmental exposures and behaviors that may influence ESCC progression along the historic timeline (beginning in the late 1800's to the 1990's) rather than periods that lie in the relatively short time window in which cancers are diagnosed. Thus, the immediate impact of historical trends on important biological processes can be explored throughout an individual's lifetime. Furthermore, most traditional APC models, even with biologically constrained age effects [15–18], generally assume proportionality of period and cohort effects with an unadjusted hazard function (which describes the age effect), an assumption that cannot fully disclose the relationship between historical period and carcinogenic exposures over time. A notable exception, however, has recently been described by Aßenmacher et al. [19] in an analysis of lung cancer mortality and radon exposures in German uranium miners.

While this analysis focuses on understanding previously recognized disparate trends in ESCC incidence by sex and race in the US, provided by the SEER registry [20], the proposed regularized method can also be used to characterize period and cohort trends associated with

other cancers to better understand the role of field cancerization and the impact of various cancer control measures over time.

## Methods

Mathematical models that capture the multistage nature of cancer, i.e. tumor initiation, clonal expansion (promotion) and transformation, have provided important insights into the stochastic dynamics and characteristic time-scales associated with the development of precursor lesions and their malignant successors [16, 18, 21, 22]. For this reason we use a modified multistage clonal expansion (MSCE) model framework to explore ESCC incidence data, following previous applications of this framework for other cancers [17, 23–28]. The basic ESCC model structure is illustrated in Fig 1 and assumes that the cancer process starts with an event that transforms part of the normal esophageal tissue into a premalignant field (field cancerization). No assumptions are made regarding the molecular nature of this field, its size and its causes. It is possible that this field occurs in response to distinct environmental insults (e.g. smoking initiation, industrial exposure, inflammation) that cause (epi)genetic defects in a large number (or group) of normal cells making them susceptible to cancerization. Cells that comprise this field are assumed at equal risk to acquire 1 or more driver mutations (with rate $\mu_1$) that allow premalignant (or "cancerized") stem cells to proliferate and to form dysplastic foci. Dysplastic cells can eventually go extinct but may also transform into one malignant daughter cell and one dysplastic daughter cell with rate $\mu_2$. Upon the appearance of a viable malignant cell, the model assumes a 5 year time window before the cancer has grown large enough to become symptomatic.

The mathematical derivation of the *hazard function*, $h(a)$, which is the rate at which cancers occur in a population of individuals who have not yet developed the cancer of interest by age $a$ (also known as *age-specific cancer incidence*), has been derived previously (e.g., [17]). Briefly, for the model depicted in Fig 1, the hazard function can be written as a convolution of a random event that leads to a premalignant field and a subsequent process representing a two- or three-stage clonal expansion model which begins with initiating mutations that occur in cells that belong to the premalignant field:

$$h(t) = \frac{\int_0^t ds\, f_{FD}(s)\, f_{MSCE}(t - s)}{1 - \int_0^t ds\, f_{FD}(s)(1 - S_{MSCE}(t - s))} \tag{1}$$

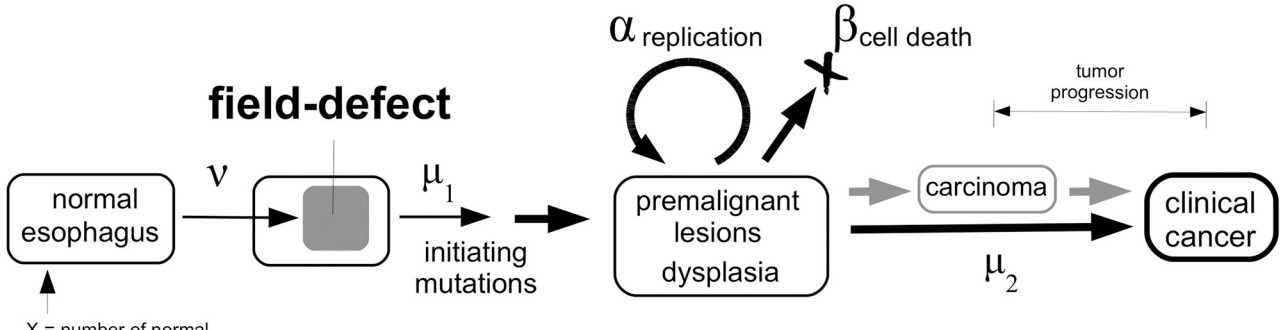

**Fig 1. Multistage clonal expansion (MSCE) model for ESCC.** Model parameters are: $\nu$ the rate of an ESCC predisposing field-defect, $\mu_1$ the mutation rate (per field) for the initiation of premalignant (dysplastic) clones, $\alpha$ the cell division rate, $\beta$ the cell death rate, and $\mu_2$ the rate of malignant transformations. A *lag-time* of 5 years is assumed for a viable malignant cell to progress to clinical cancer.

where $f_{FD}(s) = v_{FD}(s)e^{-\int_0^s dv\ v_{FD}(v)}$ is the density function for the premalignant field-defect event. Note, when the FD rate $v_{FD}$ is constant, this density function is strictly exponential. $S_{MSCE}(t-s)$ is the (tumor) survival function for the MSCE model component shown in Fig 1. Note, $f_{FD}(s)$ may be "improper" if the FD rate $v_{FD}(s)$ vanishes at some time $s < \infty$, i.e., not all individuals may ever develop the field-defect. For a two-stage clonal expansion model (i.e., a single initiating mutational event is required to induce dysplastic cell proliferation), we have

$$f_{MSCE}(u) \quad = \quad h_2(u)\ S_2(u) \tag{2}$$

with

$$h_2(u) = \frac{\mu_1}{\alpha}\frac{qp\ (e^{-qu} - e^{-pu})}{qe^{-pu} - pe^{-qu}} \tag{3}$$

and

$$S_2(u) = e^{-\int_0^u dv\ h_2(v)} = \left(\frac{q-p}{qe^{-pu} - pe^{-qu}}\right)^{\mu_1/\alpha} \tag{4}$$

with

$$
\begin{aligned}
p &= \frac{1}{2}\left(-\alpha + \beta + \mu_2 - \sqrt{(\alpha + \beta + \mu_2)^2 - 4\alpha\beta}\right) \\
q &= \frac{1}{2}\left(-\alpha + \beta + \mu_2 + \sqrt{(\alpha + \beta + \mu_2)^2 - 4\alpha\beta}\right).
\end{aligned}
\tag{5}
$$

Rather than fitting the model parameters $p$ and $q$, we use the combinations $g = -(p + q) = \alpha - \beta - \mu_2$ (net cell proliferation) and $pq = -\alpha\mu_2$ ($\approx$ malignant transformation). The premalignant cell division rate $\alpha$ is non-identifiable from incidence data. To guarantee the identifiability of all other parameters we assumed $\alpha = 17.4$/yr, the cell division rate for esophageal stem cells estimated by Tomasetti and Vogelstein [29] for normal esophagus. Analogous expressions for a convolution of a field-event and a three-stage clonal expansion process are provided in Jeon et al. [17].

## Regularized age-period-cohort (APC) model

Secular trends in cancer incidence data can be modeled using a traditional *proportional hazards* approach which assumes multiplicativity of the unadjusted hazard function (age effect) and associated period and cohort effects. A limitation of this approach is that the period effect ascribes incidence trends by year of diagnosis rather than by historical time. Here we introduce an alternative method which incorporates historic trends directly into the carcinogenesis process.

To capture historic trends that affect the incidence rate of a premalignant field-defect, $v_{FD}$, we simply allow $v_{FD}$ to depend on age $s$ and birth year. Thus, given a birth cohort $B$, $v_{FD}$ is also a function of calendar year $y$ by virtue of the relationship $y = B + s$. Fig 2 provides an illustration of our method which regularizes period effects operating on $v_{FD}$ for all calendar years $y < y_0$, where $y_0$ is a reference year independent of cohort. For $y \geq y_0$, we assume period effects associated with the field-defect to vanish, i.e. $v_{FD} = v_0$. Furthermore, assuming that the field-defect has low probability of occurring early in life, we moderate this period trend with a term that forces $v_{FD} \to 0$ below an age $s_0$, as shown in Fig 2. This assumption can be relaxed, i.e. the logarithmic term $\log(s/s_0)$ in the expression for $v_{FD}$ can be omitted (see Fig 2). However, we found significantly better fits by deviance for Blacks (both sexes) and white females with this

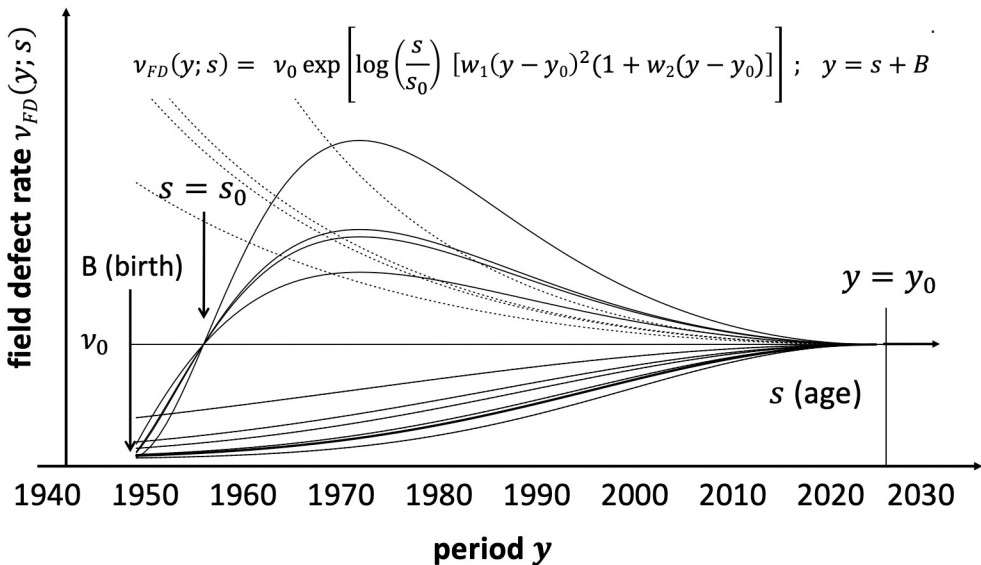

**Fig 2. Functional regularization of period (year) trends on the field-defect rate $v_{FD}$.** The parameters $s_0$, $y_0$, $w_1$, $w_2$ and $v_0$ are estimated from the data. Note, $v_{FD}$ approaches the background rate $v_0$ smoothly at the reference point $y_0$ and is assumed to remain constant for $y > y_0$. Here $s$ refers to age, $B$ to birth year, $y$ to period. For illustration, we also show some randomly generated $v_{FD}$ trajectories with the attenuating term $\log(s/s_0)$ included (solid lines) and with this term omitted (dotted lines).

term included ($p < 2 \cdot 10^{-4}$; $\tilde{\chi}^2(1\ df)$), which led to a more pronounced early increase in the field-defect rate $v_{FD}(s)$. For white males, the fit only improved marginally.

Because $v_{FD}$ may also be influenced by effects that are strongly associated with birth cohort $B$, we further introduce a multiplicative factor for $v_{FD}$ of the form $\exp[b_1(B-B_0)^2(1+b_2(B-B_0))]$, where $B_0$ is a reference birth year analogous to the period reference $y_0$. Similar to vanishing period trends for $v_{FD}$ when $y > y_0$, we also assumed vanishing cohort trends when $B > B_0$. Furthermore, we did not find significant improvements in model fits with independent estimations of $B_0$ and $y_0$. Thus, we set $B_0 = y_0$, a simplification that stipulates $v_{FD} = v_0$ when $y$ and $B$ exceed $y_0$ improving the stability of our maximum likelihood estimates (MLEs).

Finally, because general time trends may also affect rate of cancer progression, we also investigated whether the growth of dysplastic clones (within the premalignant field) shows significant trends by birth cohort. To this end, we imposed a simple linear-quadratic birth-cohort adjustment on the promotion parameter $g = \alpha - \beta - \mu_2$ of the form:

$$g = g_0 \exp[g_1(B - 1800)(1 + g_2(B - 1800))],$$

where $B$ is the birth year, as above.

## Parameter estimation

SEER9 ESCC incidence data were obtained via SEER*Stat [20] and tabulated by age and calendar year from 1975 to 2016. Birth cohorts after 1960 were excluded due to the very small number of cases. S1 Table summarizes the available number of ESCC cases and person years by sex and race for individuals born between 1900 and 1960 in 10-year increments. As in previous analyses of cancer registry data (e.g. [22]) we stratified the number of cases by age and calendar year and assumed them to be Poisson distributed with mean $\lambda = h(t) \times PYs$, where $h(t)$ is the hazard function defined in Eq (1) and $PYs$ the number of person years in a specific stratum.

Parameter estimation was performed by maximum likelihood and via Markov Chain Monte Carlo (MCMC) sampling as described in [22]. Computer code for computing the likelihood is available on https://github.com/gluebeck/ESCC-Incidence-Trends.

## Results

Here we present findings related to the main biological processes that shape the age-specific incidence of ESCC by sex and race in the US, and how parameters that control these processes are modulated by period and cohort effects. Our analysis follows previous work for colorectal cancer and esophageal adenocarcinoma, but uses the new computational approach outlined in Methods for estimating the impact of period and cohort effects on the biological model parameters.

### ESCC field cancerization

We first conducted preliminary analyses of ESCC incidence data from the SEER9 registry (years 1975-2016; birth cohorts ≤1960) using a multistage clonal expansion (MSCE) model that assumes that normal (tissue) stem cells need to acquire 2 driver mutations before they undergo stochastic clonal growth [22] (see Fig 3A). Period and cohort adjustments were performed as previously described. The best model fits (by deviance) were obtained using the regularized period and cohort adjustments acting on the first mutational event (rate $\mu_0$) for the model in which normal stem cells are considered as independent and at equal risk of generating progeny that give rise to cancer. Alternatively, we explored a model that stipulates a multicellular field-defect with rate $\nu_{FD}$ as the first event (see Fig 3B).

Maximum likelihood analyses with the two models provided similar fits to the ESCC incidence data. However, only the FD model yielded biologically plausible parameter estimates for the rates of mutations that lead to premalignant clones (see Fig 3). We arrived at this conclusion via the following reasoning: 1) in the absence of a field-defect all normal tissue stem cells are at risk of acquiring the first rate-limiting mutation with rate $\mu_0$ per cell. Assuming the human esophagus has about $10^6$ normal stem cells [29], each of which is at risk to acquire the first requisite driver mutations, we estimated $\mu_0$ to be in the range of 4.5-8.7 $10^{-11}$ per year for both sexes and races (Blacks and Whites)—mutation rates which are much smaller than the rate of sporadic DNA base substitutions, especially considering that most stem cells divide numerous times per year. More importantly, 2) estimates of the second rate-limiting mutation, $\mu_1$, were exquisitely high for all groups, i.e. >1 per year (Table 1), which suggests a large number of cells make up the field-defect since $\mu_1$ may be considered the product of a (large) number of stem cells within the field and a (small) locus-specific mutation rate per cell. Locus-specific mutation rates are known to vary with gene size and function, but are generally assumed to be in the range of $10^{-4}$ to $10^{-5}$ per year [30], depending on the rate of stem cell divisions. Thus, the field-defect, once established, appears to entail a large (>$10^4$) number of normal stem cells. Thus, a model that assumes that all normal esophageal stem cells are at equal risk to transform into a clinical esophageal malignancy did not yield plausible biological rates. However, excellent fits to incidence data along with biologically plausible rates were found when using a model where a large number of cells first establish a permissive field of activated cells that are prone to stochastic initiation of dysplasia.

### ESCC incidence by age group and calendar year

ESCC incidence rates show remarkable differences by sex and race ("Black" and "White" SEER9 populations by race code variable) and by calendar year (i.e., year of ESCC diagnosis), as seen in Fig 4. Especially striking is the ∼4-fold higher incidence of ESCC in the 1980's for

## A) Cell level initiation

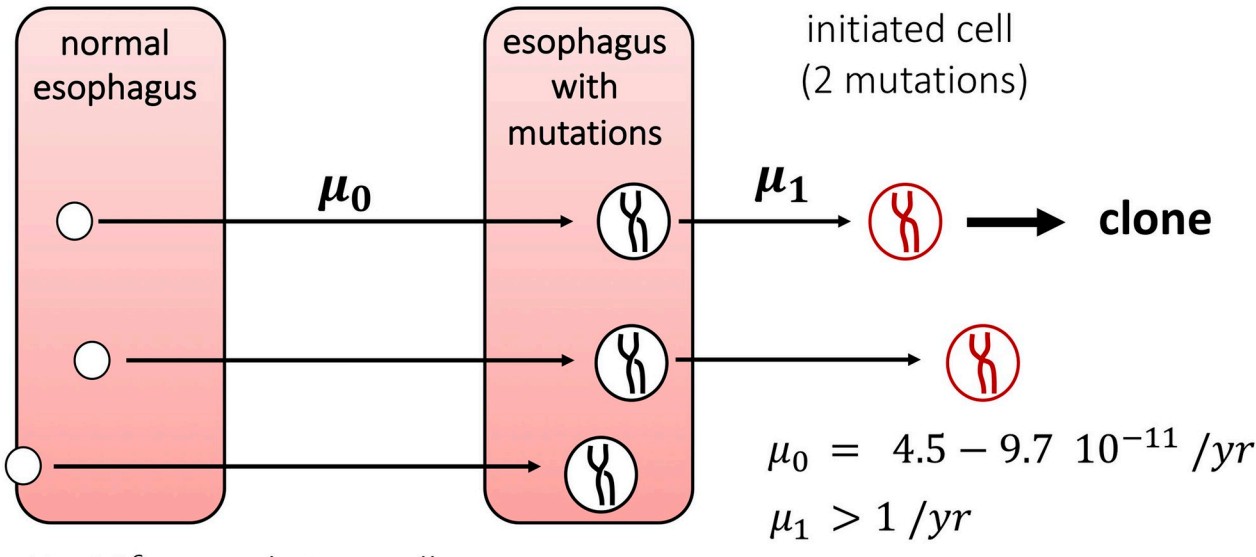

$X = 10^6$ normal stem cells

## B) Field-defect initiation

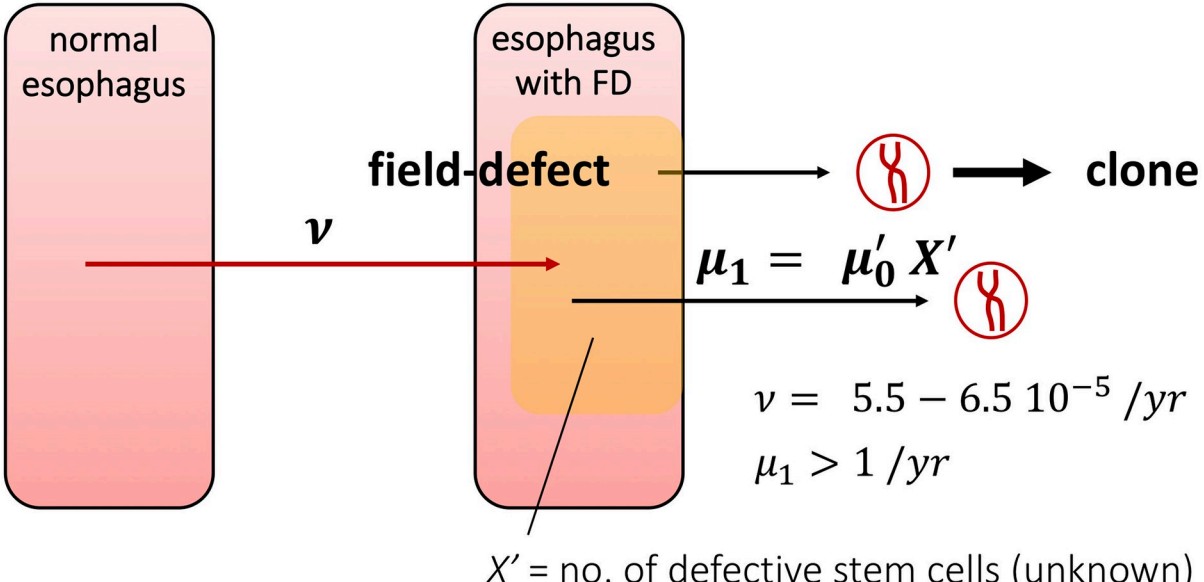

$X'$ = no. of defective stem cells (unknown)
$\mu'_0$ mutation rate of defective stem cells

**Fig 3.** A) Two-hit process for the initiation of dysplastic clones; B) Sporadic formation of field-defect followed by single hit process for the initiation of dysplastic clones. Note, only the product $\mu_1 = \mu'_0 X'$ is identifiable and represents the rate at which dysplastic clones are initiated in the defective field.

**Table 1. Parameter estimates are based on posterior samples obtained via Markov Chain Monte Carlo (MCMC) using random uniform prior distributions with boundaries that were well outside the obtained 95% confidence regions.** For sampling, we used a multivariate Metropolis-Hastings algorithm with >100, 000 samples for Males and >200, 000 samples for Females. The marginal medians listed in this table agreed well with the obtained maximum likelihood estimates and led to very similar incidence predictions (S3 Fig).

| A: marginal posterior medians (Blacks) with 95% confidence regions | | | | | | | |
|---|---|---|---|---|---|---|---|
| parameter | unit | Black Males | | | Black Females | | |
| | | **median** | lower | upper | **median** | lower | upper |
| $v_0$ (×10⁻⁵) | [1/year] | **9.54** | 8.06 | 11.4 | **5.52** | 4.38 | 7.12 |
| $\mu_1$ | [1/year] | **39.4632** | 27.5652 | 55.3248 | **15.7833** | 7.9030 | 30.8817 |
| $\mu_2$ (×10⁻⁷) | [1/year] | **0.0345** | 0.0087 | 0.1009 | **0.4878** | 0.0974 | 1.8217 |
| $g_0$ | [1/year] | **0.0249** | 0.0210 | 0.0296 | **0.0632** | 0.0443 | 0.0928 |
| $g_1$ | [1/year] | **0.0268** | 0.0248 | 0.0296 | **0.0096** | 0.0057 | 0.0136 |
| $g_2$ | [1/year] | **-0.0029** | -0.0032 | -0.0028 | **-0.0011** | -0.0022 | -0.0003 |
| $b_1$ | [1/year²] | **0.0106** | 0.0103 | 0.0111 | **0.0139** | 0.0132 | 0.0149 |
| $b_2$ | [1/year] | **0.0440** | 0.0436 | 0.0443 | **0.0610** | 0.0589 | 0.0640 |
| $w_1$ | [1/year²] | **0.0002** | 0.0002 | 0.0002 | **0.0011** | 0.0011 | 0.0012 |
| $w_2$ | [1/year] | **-0.7856** | -0.8194 | -0.7574 | **-0.2582** | -0.2779 | -0.2435 |
| $y_0$ | [year] | **1967.1696** | 1965.2571 | 1969.0098 | **1961.0000** | - | - |
| $s_0$ | [year] | **0.4954** | 0.4773 | 0.5133 | **0.3586** | 0.3150 | 0.3989 |
| B: marginal posterior medians (Whites) with 95% confidence regions | | | | | | | |
| parameter | unit | White Males | | | White Females | | |
| | | **median** | lower | upper | **median** | lower | upper |
| $v_0$ (×10⁻⁵) | [1/year] | **5.52** | 4.76 | 6.44 | **6.51** | 4.05 | 14.2 |
| $\mu_1$ | [1/year] | **4.3755** | 3.3452 | 5.5657 | **1.2988** | 0.4992 | 2.9233 |
| $\mu_2$ (×10⁻⁷) | [1/year] | **1.3273** | 0.7746 | 2.2392 | **0.3392** | 0.0602 | 1.0892 |
| $g_0$ | [1/year] | **0.1836** | 0.1686 | 0.1998 | **0.2032** | 0.1670 | 0.2488 |
| $g_1$ | [1/year] | **0.0001** | 0.0000 | 0.0010 | **0.0000** | 0.0000 | 0.0000 |
| $g_2$ | [1/year] | **0.0688** | 0.0039 | 1.3313 | **14.5845** | 3.4146 | 43.1293 |
| $b_1$ | [1/year²] | **0.0038** | 0.0033 | 0.0043 | **0.0042** | 0.0033 | 0.0053 |
| $b_2$ | [1/year] | **0.0305** | 0.0295 | 0.0315 | **0.0289** | 0.0265 | 0.0331 |
| $w_1$ | [1/year²] | **0.0006** | 0.0005 | 0.0007 | **-0.0002** | -0.0002 | -0.0001 |
| $w_2$ | [1/year] | **-0.0928** | -0.1057 | -0.0844 | **0.2786** | 0.2142 | 0.3670 |
| $y_0$ | [year] | **1963.0811** | 1961.2545 | 1966.8300 | **1961.0000** | - | - |
| $s_0$ | [year] | **1.8121** | 1.4694 | 2.2899 | **0.6246** | 0.5059 | 0.8992 |

black males and their decline to levels still much above the current incidence levels among white males. Although black males and females differ in overall levels they share rather similar curvatures for all three age groups shown in Fig 4. In contrast, the patterns for white males and females differ visibly, especially for the older age group (70+). The ESCC incidences for white males stand out from the others in that they appear to decline monotonically, showing virtually no curvature for the period 1975-2016 for which we have data, with incidence levels that approach those of white females in earlier years.

## The influence of historic period and birth-cohort on the risk of developing an esophageal field-defect

As described in Methods, the regularized convolution model for ESCC incorporates both period and cohort effects. Our model fits (Fig 5, Table 1) show that the age-specific rate at which a premalignant field-defect is induced in the esophagus prior to the development of

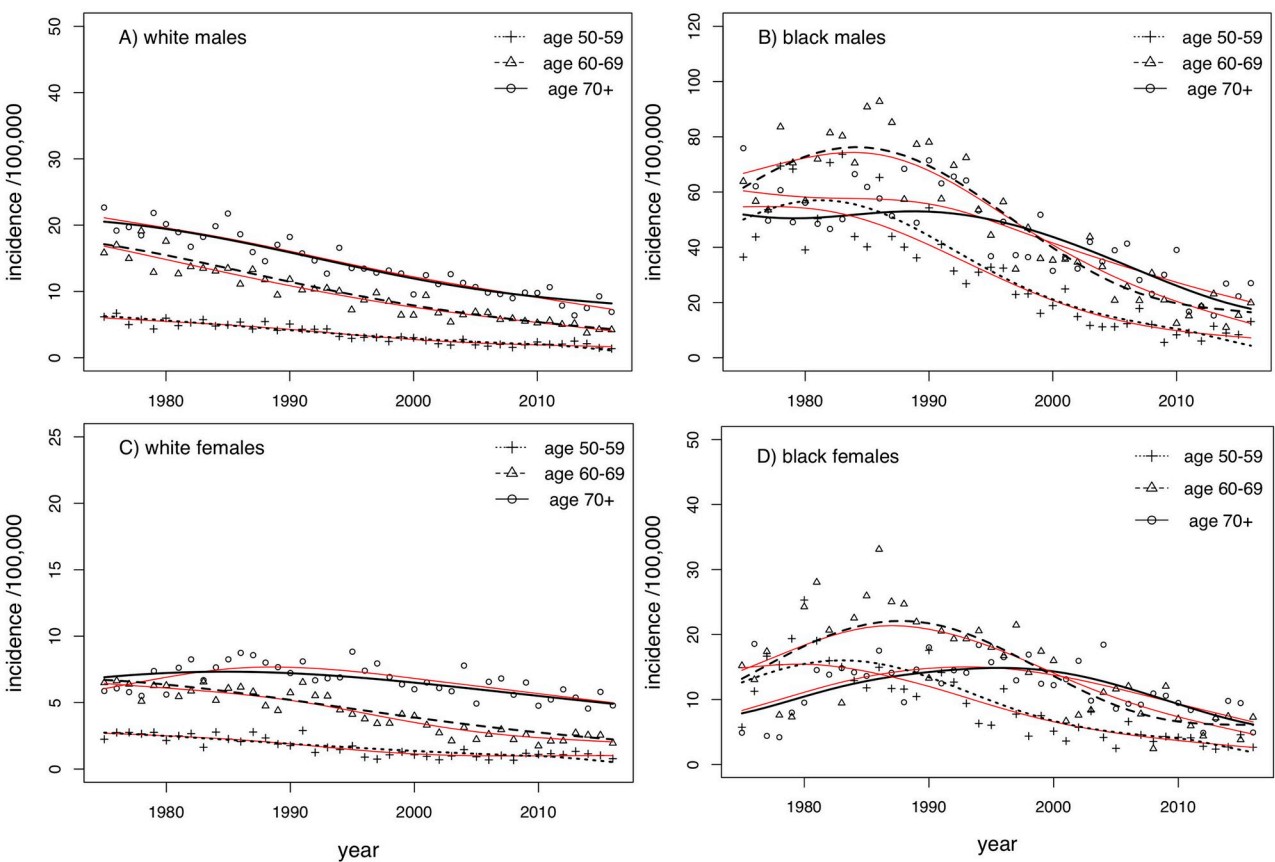

**Fig 4. ESCC incidence (data, black shapes; model fits, black lines) by sex, race and age group as indicated from 1975-2016.** For comparison, non-parametric fits using $4^{th}$-order smoothing splines are shown in red.

cancer is strongly affected by age and historical period. Fig 5 shows the predicted influence on the field cancerization rate $v_{FD}(s;B)$ as a function of period (i.e. historic time from the late 1800's to the year 1980) for fixed birth cohorts $B$, as indicated.

The emerging (estimated) time profiles for the field-defect rate $v_{FD}(s)$ show surprisingly narrow time distributions (about 5-6 years for Blacks, about 10 years for Whites) with their maxima at <10 years. These curves also differ remarkably between the races (Fig 4). For black men and women, we see very similar cohort patterns in that the presumed field-defect rate is predicted to peak sharply in the first decade of life. For Whites, the predicted rate curves are somewhat wider with their maximum between ages 5 and 10, well before smoking initiation would begin [31]. Note that the estimated period effect for $v_{FD}(s;B)$ among white males is qualitatively distinct from the other groups in that the cohort-specific maxima of $v_{FD}(s;B)$ decline monotonically as seen in Fig 5.

Fig 6 shows the estimated field-defect rate $v_{FD}(s;B)$ (panel A) and corresponding age-specific FD prevalence (panel B) for the 1920 cohort among both sexes and races (analogous plots for other cohorts is shown in S2 Fig). This comparison shows the dramatic differences in the maximal strength of the field-defect rate between the races. Beyond childhood, the predicted prevalences of the field-defect among black males is roughly 3-fold higher than for the other groups. Fig 6 also reveals that only up to 1% of Whites and black females (born in 1920) ever

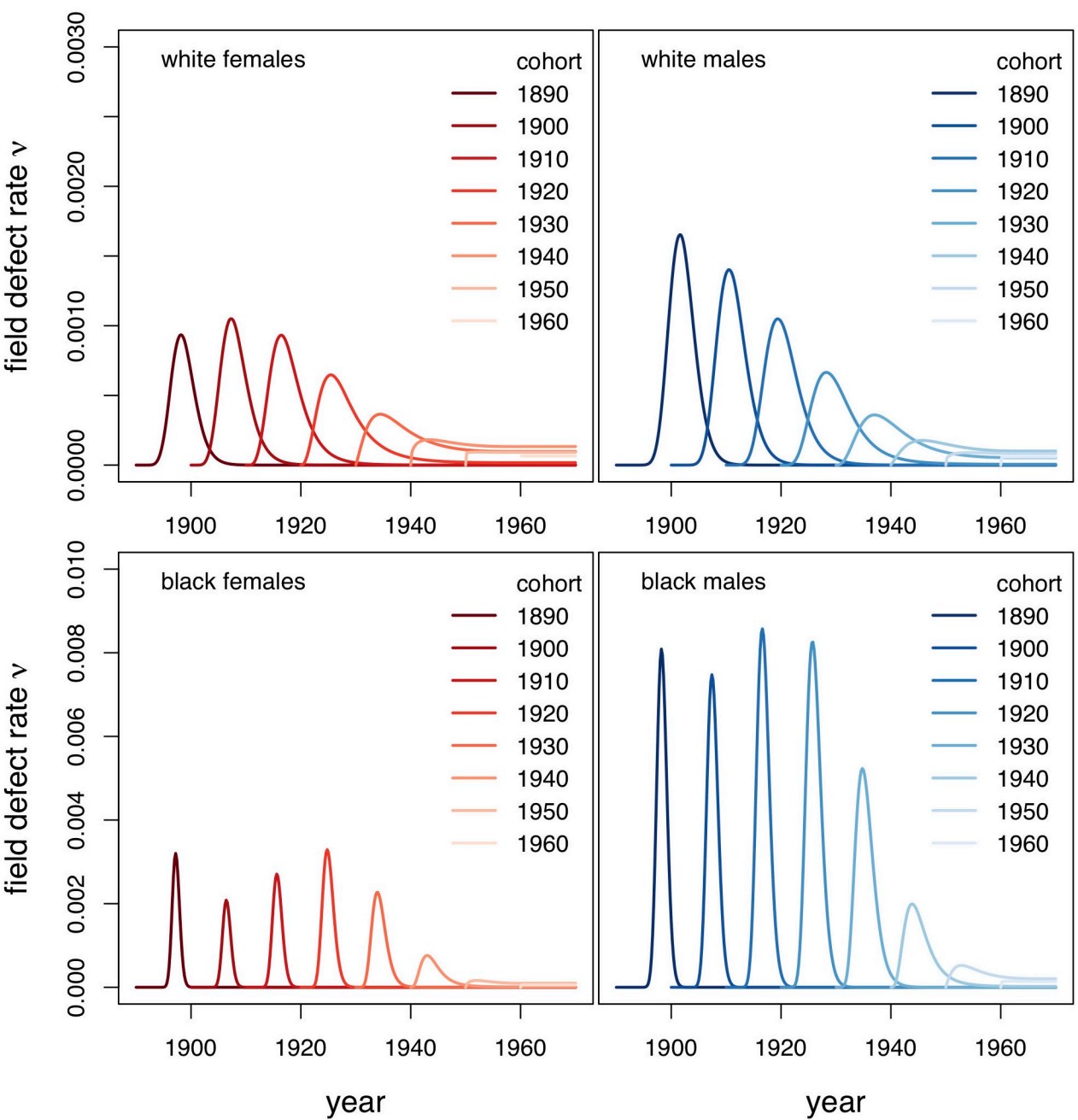

**Fig 5. Period profiles of the field-defect rate $\nu_{FD}(s;B)$ by birth cohort as indicated.**

developed the predisposing FD, while almost 3% of black males acquired this defect early in their life.

## Birth cohort effects on premalignant cell proliferation (promotion)

For Blacks (both sexes), we find significant increasing birth-cohort effects on the promotion parameter $g$ ($\tilde{\chi}^2$ (2 df) $p < 5 \cdot 10^{-5}$), assuming a null hypothesis with no trends ($g_1 = g_2 = 0$).

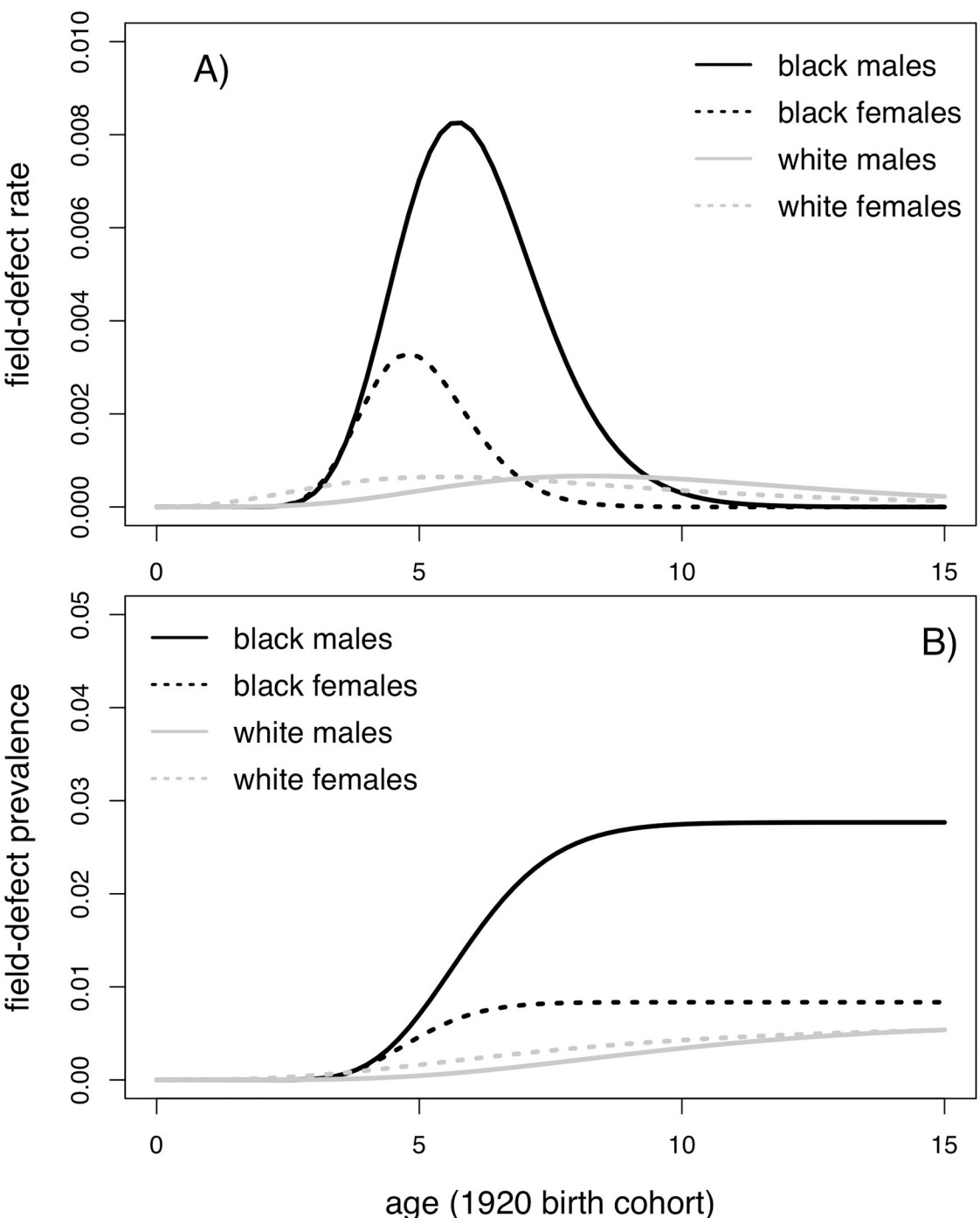

**Fig 6. Predicted field-defect rates $v_{FD}$ (panel A) and corresponding field-defect prevalences as functions of age (1920 birth cohort) by race and sex.**

Similar effects were found for white females ($p < 10^{-3}$). However, for white males, we found no significant birth cohort effect on promotion. S1 Fig shows the cell proliferation (promotion) parameter $g$ as a function of birth cohort for all 4 populations analyzed.

## Discussion

Although the origin of ESCC has previously been associated with field cancerization involving both epigenetic and genetic defects that give rise to multifocal dysplasia in esophageal and other tissues [32, 33], the clinical evidence is scant and may be confounded with the ubiquitous presence of driver mutations (*TP53, Notch1*) in normal squamous tissue [34]. Surprisingly, this analysis of ESCC incidence patterns using a mathematical model that integrates the main cellular processes involved in cancer development, suggests that the known advanced precursor to ESCC (i.e. esophageal dysplasia) arises within a premalignant field that primarily develops early in life and is under the influence of strong historic trends. Although environmental exposures, such as cigarette smoking, alcohol consumption may contribute to field formation later in life, the young-age signatures and time trends of the inferred premalignant field-defect suggest that other factors such as pediatric malnutrition and vitamin deficiencies may significantly influence ESCC incidence patterns in the US.

Whether nutrition or other factors, this study suggests that there is a critical early-childhood window of susceptibility for developing field-defects that predispose to the later development of ESCC. While our exploration of the SEER-based ESCC incidence data [20] reveal the known strong disparities of ESCC rates between Blacks and Whites [35], our results also suggest that there is a significantly higher risk for developing field-defects during childhood for Blacks than for Whites, especially among the earlier birth cohorts when nutritional or other disparities may have been more prevalent among Blacks. Other predisposing factors for ESCC include the possible impact of pathogens such as Human Papilloma Virus (HPV) [36] and interactions of the microbiome with evolving dietary patterns.

Our conjecture that the carcinogenic process for ESCC mainly begins with the development of a field-defect rather than with sporadic rare mutations in individual stem cells is based on the comparison shown in Fig 3. Both scenarios (cell-level vs FD initiation) fit the ESCC incidence patterns equally well (S2 Table) but differ in terms of the hazard function and estimated model parameters as discussed under *ESCC field cancerization*. Importantly, the sporadic (independent) stem cell mutation scenario (Fig 3A) yields mutation rates that are implausible while the field-defect scenario yields estimates that are in line with the concept of an abnormal (defective) esophageal tissue in which rare (rate-limiting) mutations lead to the initiation of dysplasia. Since mutations in *TP53* are found in virtually all ESCC [37, 38] and occur frequently in BE and EAC [39], it is plausible to assume that the first mutation in the field-defect involves *TP53* LOH, associated with loss of *TP53* tumor suppressor function, followed by a malignant transformation in response (or due) to the loss of *TP53* control consistent with Knudson's two-hit model [40]. Even so, it is possible that some ESCC arise from sporadic initiating mutations in normal esophageal tissue without involving a FD. However, the presence of a non-FD pathway (or pathways) to ESCC should tend to wash out the field-defect rate which we infer to be rather concentrated temporally.

Although the molecular nature of the conjectured field-defect for ESCC is still unclear, it has been associated with epigenetic alterations associated with smoking and alcohol consumption confounded by other factors such as low BMI and poor diet [41]. Because our data lack covariate information on such factors, the clinical relevance of our findings is limited. Still, we expect this modeling approach to be useful when it comes to cancers with well recognized

field-defects, such as lung cancer, colorectal cancer, and esophageal adenocarcinoma (EAC) for which specific field-defects have been described in the literature [42].

In summary, our findings suggest the disparate demographic patterns observed in the US may not be wholly attributable to smoking and alcohol, but may also reflect other factors acting early in life. This conclusion is supported by the similarity of recently re-assessed historic smoking initiation and prevalence distributions in the US between Whites and Blacks [31]. Modest differences in smoking and alcohol use alone may therefore not fully explain the high degree of racial divergence and recent converging trends in ESCC incidence between Whites and Blacks. Environmental exposures, such as cigarette smoking, alcohol and industrial exposures may well contribute to field formation later in life (and increased promotion of precursor lesions), however the young-age signatures and time trends of the inferred premalignant field-defect suggest that other factors such as pediatric malnutrition, iron deficiency anemia (IDA) and deficiencies of some B vitamins (e.g. riboflavin, thiamin) may more significantly be involved in shaping the historic ESCC incidence patterns in the US [43, 44].

## Supporting information

**S1 Table. SEER9 1975-2016 data summary.** Listed are the total number of person years (PYs) and the number of ESCC cases stratified by 10-year birth cohorts up to 1960.
(XLSX)

**S2 Table. Deviances.** Listed are Poisson deviances for the models that we fitted to each of 2043 single year age-period strata for the 4 populations studied. This Table shows that, with the exception of males, the regularized FD model is preferable to the saturated Poisson model. The saturated model has p-values of 0.034 and 0.07 for black males and white males, respectively. Thus, the FD model cannot be soundly rejected on statistical grounds. The table also confirms that there is generally little difference in the deviances between non-FD (stem cell independence) and FD models.
(XLSX)

**S1 Fig. Promotion.** Estimated cell proliferation parameter $g$ as a function of birth cohort.
(TIF)

**S2 Fig. ESCC field-defect rates.** For the 1930, 1940 and 1950 birth cohort for both sexes and races.
(TIF)

**S3 Fig. Comparison of MLE- and MCMC-generated ESCC incidence predictions.** ESCC incidence between 1975 and 2016 for black males and females at age 65. Similar results for Whites (not shown). Bands indicate 95% MCMC-based credibility regions.
(TIF)

## Author Contributions

**Conceptualization:** Georg E. Luebeck.

**Data curation:** William D. Hazelton.

**Formal analysis:** Georg E. Luebeck.

**Funding acquisition:** Georg E. Luebeck, Thomas L. Vaughan, William D. Hazelton.

**Methodology:** Georg E. Luebeck, Kit Curtius, William D. Hazelton.

**Project administration:** Georg E. Luebeck.

**Software:** Georg E. Luebeck, William D. Hazelton.

**Supervision:** Georg E. Luebeck.

**Visualization:** Thomas L. Vaughan, Kit Curtius, William D. Hazelton.

**Writing – original draft:** Georg E. Luebeck.

**Writing – review & editing:** Thomas L. Vaughan, Kit Curtius, William D. Hazelton.

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
