## [Decision Letter · Decision Letter 0]

26 Feb 2021

Dear Dr. Luebeck,

Thank you very much for submitting your manuscript "Modeling historic incidence trends implies early field cancerization in esophageal squamous cell carcinoma" for consideration at PLOS Computational Biology. As with all papers reviewed by the journal, your manuscript was reviewed by members of the editorial board and by several independent reviewers. The reviewers appreciated the attention to an important topic. Based on the reviews, we are likely to accept this manuscript for publication, providing that you modify the manuscript according to the review recommendations.

Sincerely,

Dominik Wodarz

Associate Editor

PLOS Computational Biology

Florian Markowetz

Deputy Editor

PLOS Computational Biology

[LINK]

Reviewer's Responses to Questions

**Comments to the Authors:**

Reviewer #1: Review is uploaded as attachment.

Reviewer #2: This manuscript describes an analysis of esophageal squamous cell carcinoma incidence trends in the US by race and sex. It applies the multistage carcinogenesis model that provides a good fit to the data. In addition, the model uses a biological model that provides insights into mechanisms that may be at work at the cellular level. This suggests that there may be factors or exposures that come into play early in life that may be the cause of the very different trends that are apparent by race and sex.

The conclusions from this analysis offer interesting aspects of the etiology of this disease that are useful to pursue for a better understanding of the etiology. Having said that, an important limitation of this work is that there are still aspects of the problem that are not resolved. While various versions of the multistage carcinogenesis model seem to work well for many cancer sites, it is difficult to verify all of the details at the cellular level. In addition, the data are not available for identifying the specific factors that are at play in giving rise to these different trends. For example, the results suggest that the field-defect rate has changed for cohorts in ways the differ by race and sex (Figure 5) but at this time we do not have the data that would point to the reasons for these changes.

A puzzling aspect of this manuscript is the claim that this method disentangles the identifiability problem in age-period-cohort models (p2. l.20). This is a statistical model which is linear with additive contributions for each of the time elements. A log transformation is commonly used for disease rates, so this is equivalent to a multiplicative model, as stated by the authors. Period is the sum of age and cohort, so in a way the model considered in this manuscript includes all these temporal elements. However, in the statistical model one is trying to estimate effects identified with each of the temporal elements. In this development, the progression of disease is considered for each cohort, so that the underlying effects are associated with age and cohort. It not clear what contribution is uniquely attributable to period. Such an effect might affect all cohorts at a particular time, but there is nothing like this in the model, or at least this is not obvious. Period appears to be identified with “regularization”, but it would be helpful to explain more fully what this is and how it comes into the overall model. Figure 2 needs more detail, such as a scale for the vertical axis. What do the lines in the graph represent? Are these for different values of a and b? Do we have such a graph for each B, for example?

On the one hand, the relationship with APC seems irrelevant because this a biological model that provides insight into disease trends. However, it could be an interesting aside if it does fit directly into the framework of an APC model. If it does not, I do not think that this would detract from the value of this contribution.

**Have all data underlying the figures and results presented in the manuscript been provided?**

Reviewer #1: Yes

Reviewer #2: **No: **While the data have not been provided, they are available from public websites.

PLOS authors have the option to publish the peer review history of their article (what does this mean?). If published, this will include your full peer review and any attached files.

Reviewer #1: No

Reviewer #2: No

Figure Files:

Data Requirements:

Reproducibility:

References:

---

## [Editor Report · Decision Letter 1]

13 Apr 2021

Dear Dr. Luebeck,

We are pleased to inform you that your manuscript 'Modeling historic incidence trends implies early field cancerization in esophageal squamous cell carcinoma' has been provisionally accepted for publication in PLOS Computational Biology.

Best regards,

Dominik Wodarz

Associate Editor

PLOS Computational Biology

Florian Markowetz

Deputy Editor

PLOS Computational Biology

---

## [Editor Report · Acceptance letter]

27 Apr 2021

PCOMPBIOL-D-20-02290R1 

Modeling historic incidence trends implies early field cancerization in esophageal squamous cell carcinoma

Dear Dr Luebeck,

I am pleased to inform you that your manuscript has been formally accepted for publication in PLOS Computational Biology. Your manuscript is now with our production department and you will be notified of the publication date in due course.

With kind regards,

Andrea Szabo
